# Grain Dehydration Characteristics of Old and Modern Maize Hybrids and Their Response to Different Planting Densities

Huaihuai Guo [1], Xiaofang Yu [1,*], Julin Gao [1,*], Daling Ma [1], Shuping Hu [2] and Xian Wang [3]

[1] College of Agronomy, Inner Mongolia Agricultural University, Hohhot 010019, China; 18447053537@163.com (H.G.); madaling@sina.com (D.M.)
[2] Vocational and Technical College, Inner Mongolia Agricultural University, Salaqi 014109, China; bthsp88@163.com
[3] College of Horticulture and Plant Protection, Inner Mongolia Agricultural University, Hohhot 010019, China; wx13664771913@163.com
* Correspondence: yuxiaofang75@163.com (X.Y.); nmgaojulin@163.com (J.G.); Tel.: +86-136-7482-7018 (X.Y.); +86-137-0475-3317 (J.G.)

**Abstract:** High grain water content at harvest stage is the main limiting factor for realizing mechanized maize grain harvest in China. Under the background of yield gain by density increase, it is necessary to clarify the effect of planting density increase on maize grain dehydration characteristics, which would provide theoretical support for realizing mechanized grain harvest under high planting density conditions. Therefore, this study selects five main hybrids, ZD2, DY13, YD13, XY335, and DH618, released in different eras that were widely promoted in Inner Mongolia from the 1970s to 2010s. The experiment was conducted in the Chilechuan Modern Agriculture Expo Park, Tumed Right Banner, Baotou city, Inner Mongolia, in 2018 and 2019. Under the three densities of 45,000 plants ha$^{-1}$ (low density), 75,000 plants ha$^{-1}$ (medium density) and 105,000 plants ha$^{-1}$ (high density), the indexes of grain dehydration, leaf stay-green, bract and cob dehydration of the different maize hybrids were measured and analyzed. The results show that MCpm (moisture content at physiological maturity) of hybrids in the 1970s and 1990s was significantly reduced by 1.57 and 1.14 percentage points, respectively, and MCh (moisture content at harvest time) in the harvest period of hybrids in the 1970s was significantly reduced by 0.99 percentage points, from a low to medium density. The GDRbm (rate of grain dehydration before maturation) and the GDRam (rate of grain dehydration after maturation) showed an increasing trend from a low to medium density. From a medium to high density, the MCpm from the 1980s to 2000s could be significantly reduced by 1.78, 1.53 and 1.88 percentage points; the MCh from the 1980s could be significantly reduced by 1.77 percentage points; and the GDRbm from the 1970s was significantly increased by 0.101%/d, but the improvement of GDRam was not significant. With the planting density increase, the decreased ratios of relative GLAD (green leaf area duration) and leaf SPAD (soil and plant analyzer development) per plant of old maize hybrids were more than that of modern maize hybrids, which promoted the decrease in grain water content and the rate increase in grain dehydration for old maize hybrids. There was a direct positive correlation between the bract and grain dehydration rates, but the cob dehydration rate had no significant effect on the grain dehydration rate. With the increase in planting density, the relative GLAD and leaf SPAD values of plants decreased, and the stay-green of plants worsened, and a significant increase in the dehydration rate of bracts in old and modern eras was an important reason for the decrease in grain moisture content and increase in dehydration rate.

**Keywords:** maize; different eras; planting density; dehydration characteristics

## 1. Introduction

Maize (*Zea mays* L.) is an important food crop in China, and its yield ranks first among the four domestic food crops, contributing more than 80% to China's food yield [1]. Therefore, maize production is of strategic significance to ensure China's food security. In recent years, the maize industry is facing the transformation and upgrading of the

"transforming mode and adjusting structure", and efforts to improve the yield per unit area and change the mode of production are the direction and task of future research. Increasing planting density is one of the key technologies to exploit the yield potential of maize [2–4]. One of the reasons for the continuous improvement of maize yield is the reasonable increase in planting density [5,6]. In addition, the mechanized grain harvest of maize is the key to realize the whole mechanization of maize production and change the production mode, and is also the inevitable development direction of the maize harvesting mode in China [7,8]. However, at present, in the process of mechanized grain harvest in China, there are still certain problems, such as high grain damage rate, large harvest loss and low harvest quality, due to the high grain moisture content, which greatly restrict the large-scale promotion and application of the whole-process mechanization technology of maize and affect the harvest quality and production efficiency of maize [9]. Increasing planting density not only improves maize yield, but also significantly affects grain filling and the dehydration process [10]. Under the background that increasing planting density has become an effective way to increase yield, the effect of increasing planting density on grain dehydration characteristics of maize is related to whether the direct harvesting of mechanized grains after increasing planting density is affected. Therefore, it is necessary to conduct relevant theoretical studies to clarify the effect of planting density on grain dehydration characteristics.

Maize grain water content is affected by genotype, cultivation practices and environmental conditions [11]. Among them, increasing planting density has been shown to play a significant role on maize grain dehydration rates [12]. Studies have shown that, with the increase in planting density, the water content of grain decreases at harvest [13]. Widdicombe et al. [14] also observed that the grain dehydration rate of early maturing and late-maturing hybrids was slightly accelerated with the increase in planting density in the North American maize belts. However, Long et al. [15] showed that the increase in planting density would lead to the decrease in maize dehydration rates. It had also been reported that density in the range of 45,000 to 67,500 plants ha$^{-1}$ has little effect on grain moisture [16]. Therefore, there was still no consistent conclusion about the effect of increasing planting density on maize grain dehydration characteristics.

The moisture flux of water lost from grains to the atmosphere was affected by the characteristics of pericarp, bracts and cobs [17,18]. The moisture content of grains was significantly positively correlated with the moisture content of bracts and cobs [19]. If there are many layers, a large area and high water content of bracts, the dehydration rate of seeds is slow; if the bracts are few and thin, and the covering is loose and short, the water loss rate of seeds is rapid [20]. At the same time, small and thin cobs were also conducive to grain dehydration [21]. In addition, Hicks et al. [22] showed that the defoliation of plants or other treatments to reduce the green leaf area (such as mechanical damage, diseases and insects, livestock damage, etc.) would accelerate the rate of grain dehydration in physiologically mature descendants.

Due to the different production requirements and environmental conditions in different eras, there were great differences in the grain dehydration characteristics of selected maize hybrids. For example, China's breeding of maize hybrids in the 1990s, due to the one-sided pursuit of high yields, led to hybrids of the grouting rate not being high or the duration of the high-speed grouting being shorter, the grain filling stage drying time being shorter, and to the maize grain harvest when moisture content was too high. However, there were some early hybrids, such as Zhongdan 2 (1970s), with a high kernel and cob dehydration rate that also had the necessary excellent characteristics for the mechanical grain harvesting of maize [23]. In order to clarify the law of maize grain dehydration characteristics to planting density, this study selects five representative maize hybrids to analyze the commonness and individual differences of grain dehydration characteristics before and after physiological maturity under low, medium and high densities, respectively. To explore the law of maize grain dehydration rate with the change in planting density, and to clarify the difference of dehydration characteristics between old and modern hybrids

with the change in planting density. The research results can provide a theoretical reference for breeding maize hybrids with density resistance and mechanized grain harvest.

## 2. Materials and Methods

### 2.1. Site Description

The experiment was conducted in the China Chilechuan Modern Agriculture Expo Park (40°28′28″ N, 110°29′5″ E) in Tumed Right Banner, Baotou City, Inner Mongolia, located in the Tumed Plain, with a continental semi-arid monsoon climate. In 2018, the average temperature was 20.1 °C and the rainfall was 464.7 mm during the maize growth period; the average temperature was 20.7 °C and the rainfall was 370.9 mm in 2019. The soil type of the test field was sandy loam, and the soil basic fertility of the test area during the test is shown in Table 1. Additionally, the main meteorological factors during the growth period in the experimental area are shown in Figure 1.

**Table 1.** Soil basic fertility in the test area.

| Years | Organic Matter (g/kg) | Total Nitrogen (g/kg) | Alkaline N (mg/kg) | Alkaline K (mg/kg) | Alkaline P (mg/kg) | PH |
|---|---|---|---|---|---|---|
| 2018 | 19.63 | 1.48 | 78.18 | 155.19 | 17.06 | 7.95 |
| 2019 | 17.74 | 1.31 | 79.31 | 144 | 18.45 | 8.5 |

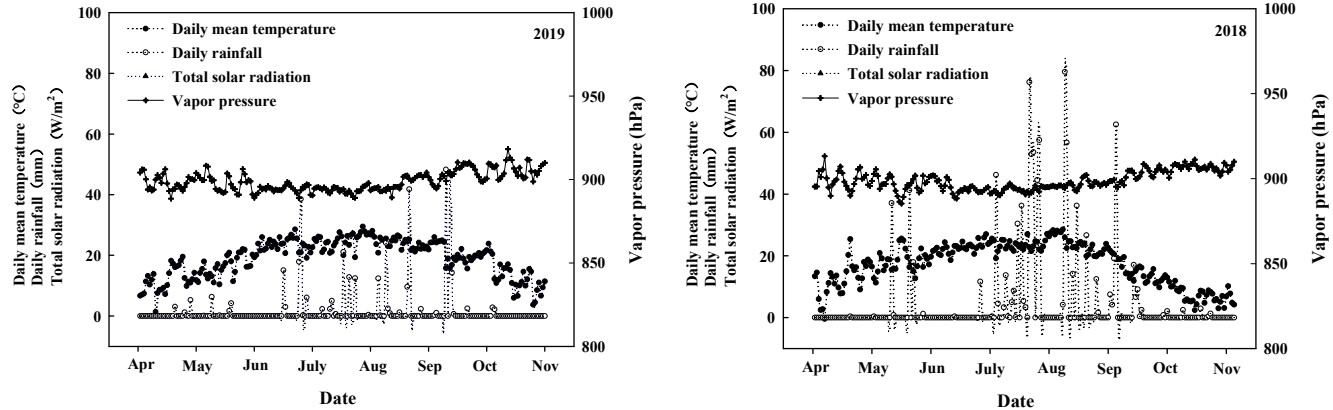

**Figure 1.** Main meteorological factors during the growth period in the experimental area.

### 2.2. Experimental Design and Field Management

The experiment adopted a two-factor split-plot design. Planting densities were the main plot; three planting densities were designed, which were a low planting density of 45,000 plants ha$^{-1}$ (D1), medium planting density of 75,000 plants ha$^{-1}$ (D2) and high planting density of 105,000 plants ha$^{-1}$ (D3), respectively; the row spacing of all planting densities were 60 cm, and the planting spacing were 37.06 cm, 22.23 cm and 15.88 cm, respectively. The sub-plots were hybrids. Five main hybrids released in different eras were selected, Zhongdan 2 (1970s, ZD2), Danyu 13 (1980s, DY13), Yedan 13 (1990s, YD13), Xianyu 335 (2000s, XY335) and Denghai 618 (2010s, DH618), which all showed medium–late maturity in Inner Mongolia, and the pedigree of hybrids is shown in Table 2. These hybrids were sold in Chinese markets, purchased as test materials, and were widely planted in the corresponding eras. There was a total of 15 treatment combinations with 3 replicates, and 45 plots in total. Each plot area was 8 m × 6 m with 9 rows. Pure N: 225 kg ha$^{-1}$, P$_2$O$_5$: 105 kg ha$^{-1}$ and K$_2$O: 45 kg ha$^{-1}$ were applied as basal fertilizers at seeding. N fertilizer was applied with water in a ratio of 3:6:1 at the stage at V6 (sixth leaf), V12 (twelfth leaf) and R2 (blister), respectively. The plots were irrigated four times during the growth period (seeding stage, V12, R1 (silking) and R2). Other field management is the same as the general field production.

**Table 2.** Pedigree table of tested hybrids.

| Hybrids | Eras | Institution Developing the Hybrid | Male Parent | Female Parent |
|---|---|---|---|---|
| ZD2 | 1970s | Chinese AAS, Beijing, China | Zi330 | Mo17 |
| DY13 | 1980s | Dandong AAS of Liaoning Province, Dandong, China | E28 | Mo17 |
| YD13 | 1990s | Laizhou AAS of Shandong Province, Laizhou, China | Dan340 | Ye478 |
| XY335 | 2000s | The Tieling Pioneer limited company, Tieling, China | PH4CV | PH6WC |
| DH618 | 2010s | Shandong Denghai Seeds Co., Ltd, Laizhou, China | DH392 | 521 |

*2.3. Measurement Indicators and Methods*

2.3.1. Grain Dehydration Characteristics

According to the growth period of different treatments, the investigation and sampling were started 15 days after silking, and samples were obtained every 3 days. Three ears were randomly selected from each treatment as 3 biological replicates.

The maize grain moisture content was measured by the oven method (TODM). First, 100 grains from the middle of the ear were obtained to measure their fresh weight, then the dry weight was weighed after being dried in the oven at 80 °C, and the corresponding grain moisture content was obtained. Determination of physiological maturity: grain dry weight usually reaches its maximum and kernels are said to be physiologically mature; physiological maturity occurs shortly after the kernel milk line disappears and just before the kernel black layer forms at the tip of the kernels. Determination of harvest date: 15 days after physiological maturity was used as the common harvest standard.

The grain moisture content at the physiological maturity stage (MCpm) (%) = (fresh weight at physiological maturity stage—dry weight at physiological maturity stage)/fresh weight at physiological maturity stage × 100.

The grain moisture content at harvest time (MCh) (%) = (fresh grain weight at harvest time—dry grain weight at harvest time)/fresh grain weight at harvest time × 100.

Grain dehydration rate before physiological ripening (GDRbm) (%/d) = (moisture content of grains sampled for the first time—moisture content of grains at physiological ripening)/interval days.

Grain dehydration rate after physiological maturity (GDRam) (%/d) = (grain moisture content at physiological maturity—grain moisture content at harvest)/interval days.

2.3.2. Dehydration Characteristics of Ears of Maize

The sampling period was the same as the grain. Three ears were randomly selected from each treatment, and the fresh weight of the bracts and cobs was measured; after drying at 80 °C, the dry weight was weighed. After harvesting, the water content and dehydration rate were calculated by sampling for the last time.

Dehydration rate of the bracts and cobs before physiological maturity (BDRbm, CDRbm) (%/d) = (water content of bracts and cobs at first sampling—water content of bracts and cobs at physiological maturity)/interval days.

Dehydration rate of bract and cob after physiological maturity (BDRam, CDRam) (%/d) = (water content of bract and cob at physiological maturity—water content of harvested bract and cob)/interval days.

2.3.3. Duration of Relative Green Leaf Area after Anthesis (Relative GLAD)

Five plants were randomly selected from the middle of each plot, and the green leaf area of every single plant with different treatments was investigated at a silking stage, 15d, 30d, 45d and 60d after the silking, respectively. The length and width coefficient method was used to calculate the leaf areas (expanded) = 0.75 × Length × width and leaf areas

(unexpanded) = 0.5 × Length × width. The total area under the curve was expressed as absolute GLAD [24,25] by plotting the time of the green leaf area of maize in the five periods after silking, namely, the cumulative number of green leaf areas per plant after flowering. Then, the relative GLAD [24,25] can be obtained by dividing the absolute GLAD by the green leaf area at flowering, which can be used to measure the greenness of the hybrid itself.

$$\text{Absolute GLAD} = \left( \int_0^n y = ax^3 + bx^2 + cx + y_0 \right) / n$$

Relative GLAD = absolute GLAD/green leaf area during silking

### 2.3.4. Chlorophyll Relative Content (SPAD)

At the silking stage, 15d, 30d, 45d and 60d after silking, the relative chlorophyll content (SPAD value) of the middle-upper surface of maize leaves at ear position (10 points per leaf and 3 plants per treatment) was measured by using the hand-held SPAD-502 chlorophyll meter (Konica Minolta, Tokyo, Japan).

### 2.4. Statistical Analysis

As the trends of all indicators in 2018 and 2019 are consistent, the data analysis in this paper was calculated using the average data for two years [26]. Statistical analysis was performed using Microsoft Excel 2016 (Microsoft, Inc., Redmond, WA, USA). SAS 9.4 statistical software (SAS Institute Inc., CA, USA) was used to test the effect of the main factors. LSD was used for the significance test. Additionally, Sigma Plot 12.5 (Systat Software, Inc., San Jose, CA, USA) was used for plotting.

## 3. Results

### 3.1. Effects of Planting Density on Grain Moisture Content at Physiological Maturity and Harvest Stage of Different Hybrids

According to the results of variance analysis in Tables 3 and 4, there were significant differences in the grain moisture content among the hybrids, density and hybrids-by-density interactions at the physiological maturity stage. The interaction of hybrids, density and hybrids × density had a significant effect on the grain moisture content at harvest time.

**Table 3.** Variance analysis of grain water content at physiological maturity stage.

| Source of Variation | 2018 | | 2019 | |
|---|---|---|---|---|
| | DF | Mean Square | DF | Mean Square |
| Block | 2 | 0.378 | 2 | 0.082 |
| Density(D) | 2 | 16.406 ** | 2 | 23.494 ** |
| Plot error | 4 | 0.196 | 4 | 0.849 * |
| Hybrids(H) | 4 | 6.166 ** | 4 | 15.009 ** |
| D*H | 8 | 0.710 ** | 8 | 0.521 |
| Subplot error | 24 | 0.186 | 24 | 0.247 |

Different letters indicate statistically significant differences at the $p < 0.05$ level. * Significant at $p < 0.05$, ** significant at $p < 0.01$, and if there is no sign, it is not significant.

As can be seen in Table 5, the grain moisture content of maize at physiological maturity and the harvest stage first increased and then decreased with the development of the hybrids in each era, reaching the highest value in the 1990s. The grain moisture content in the physiological maturity period varied from 29.79 to 34.46%, and in the harvest period from 23.87 to 26.83%. The grain moisture content in the 2000s and 2010s was lower than that in the 1970s–1990s.

**Table 4.** Variance analysis of grain water content at harvest stage.

| Source of Variation | 2018 | | 2019 | |
|---|---|---|---|---|
| | DF | Mean Square | DF | Mean Square |
| Block | 2 | 0.176 | 2 | 0.082 |
| Density(D) | 2 | 5.324 ** | 2 | 23.494 ** |
| Plot error | 4 | 0.511 | 4 | 0.849 |
| Hybrids(H) | 4 | 5.282 ** | 4 | 15.010 ** |
| D*H | 8 | 0.654 * | 8 | 0.521 * |
| Subplot error | 24 | 1.970 | 24 | 0.250 |

Different letters indicate statistically significant differences at the $p < 0.05$ level. * Significant at $p < 0.05$, ** significant at $p < 0.01$, and if there is no sign, it is not significant.

**Table 5.** Grain water content of different hybrids at physiological maturity and harvest stage under different densities in 2018 and 2019.

| Hybrids | Plant Density | 2018 | | 2019 | |
|---|---|---|---|---|---|
| | | MCpm (%) | MCh (%) | MCpm (%) | MCh (%) |
| ZD2 | D1 | 33.50 b | 25.56 ab | 33.07 bc | 27.10 b |
| | D2 | 32.20 d | 24.05 def | 31.24 de | 26.62 bc |
| | D3 | 30.45 g | 23.40 fg | 30.90 e | 24.98 def |
| DY13 | D1 | 33.65 b | 25.21 abc | 33.99 ab | 27.49 ab |
| | D2 | 32.95 bc | 25.81 a | 33.64 b | 26.87 b |
| | D3 | 31.47 de | 24.04 def | 31.56 de | 25.66 cde |
| YD13 | D1 | 34.52 a | 25.46 ab | 34.80 a | 28.19 a |
| | D2 | 33.15 b | 25.40 ab | 33.89 ab | 28.13 a |
| | D3 | 31.82 de | 24.67 bcd | 32.17 cd | 27.32 ab |
| XY335 | D1 | 32.28 cd | 24.39 cde | 31.90 de | 25.89 cd |
| | D2 | 32.15 cd | 24.03 def | 31.18 de | 25.76 cde |
| | D3 | 30.63 fg | 23.34 fg | 28.95 f | 24.50 f |
| DH618 | D1 | 31.29 ef | 23.39 fg | 31.79 de | 25.56 de |
| | D2 | 31.27 ef | 23.67 efg | 31.43 de | 25.14 def |
| | D3 | 30.59 fg | 22.95 g | 29.67 f | 24.79 ef |

MCpm: moisture content at physiological maturity; MCh: moisture content at harvest time. Different letters on the graph indicate significant differences at $p < 0.05$.

During physiological maturity, the grain moisture content of five hybrids decreased by 1.57, 0.53, 1.14, 0.43 and 0.19 percentage points successively from a low to medium density. The grain moisture content of hybrids in the 1970s and 1990s showed significant differences between the planting density ($p \leq 0.05$). There was no significant difference between the 1980s, 2000s and 2010s hybrids. From medium to high density, the grain moisture content continued to decrease, and the five hybrids decreased by 1.04, 1.78, 1.53, 1.88 and 1.22 percentage points in turn. The difference of grain moisture content between planting densities in the 1980s–2000s hybrids was significant, while the difference between the 1970s and 2010s hybrids was not significant. The results show that the grain water content in the 2010s is less affected by density, the grain water content in the 1900s is sensitive to planting density, the grain water content in the 1980s and 2000s is more affected by high density and the grain water content in the 1970s is less affected by low density.

At the harvest stage, the grain moisture content decreased by 0.99, 0.01, 0.06, 0.25 and 0.07 percentage points successively from a low to medium density. Only the grain moisture content of the 1970s hybrids had significant differences in the planting density, while other hybrids had no significant differences. From medium to high density, the grain water content decreased by 0.65, 1.77, 0.73, 0.69 and 0.72 percentage points successively. Only the 1980s hybrids had significant differences in the grain water content between planting densities, while other hybrids had no significant differences. The results show that hybrids

in the 1970s are significantly affected by low density, 1980s are significantly affected by high density and grain water content in the 1990s to 2010s is slightly affected by density.

### 3.2. Changes in Grain Dehydration Rate before and after Physiological Maturity at Different Planting Densities

According to the results of the variance analysis in Tables 6 and 7, it can be found that the interaction of density, hybrids and density × hybrids has a significant influence on the grain dehydration rate before physiological maturity. Density and hybrids had a significant effect on the dehydration rate of the grain after physiological maturity.

**Table 6.** ANOVA analysis of seed dehydration rate before physiological maturity.

| Source of Variation | 2018 | | 2019 | |
|---|---|---|---|---|
| | DF | Mean Square | DF | Mean Square |
| Block | 2 | 0.001 | 2 | 0.003 |
| Density(D) | 2 | 0.008 ** | 2 | 0.048 ** |
| Plot error | 4 | 0.000 | 4 | 0.000 |
| Hybrids(H) | 4 | 0.004 ** | 4 | 0.013 ** |
| D*H | 8 | 0.001 | 8 | 0.003 * |
| Subplot error | 24 | 0.001 | 24 | 0.001 |

Different letters indicate statistically significant differences at the $p < 0.05$ level. * Significant at $p < 0.05$, ** significant at $p < 0.01$, and if there is no sign, it is not significant.

**Table 7.** ANOVA analysis of seed dehydration rate after physiological maturity.

| Source of variation | 2018 | | 2019 | |
|---|---|---|---|---|
| | DF | Mean Square | DF | Mean Square |
| Block | 2 | 0.000 | 2 | 0.009 |
| Density(D) | 2 | 0.005 ** | 2 | 0.030 ** |
| Plot error | 4 | 0.001 | 4 | 0.000 |
| Hybrids(H) | 4 | 0.022 ** | 4 | 0.016 * |
| D*H | 8 | 0.000 | 8 | 0.008 |
| Subplot error | 24 | 0.001 | 24 | 0.004 |

Different letters indicate statistically significant differences at the $p < 0.05$ level. * Significant at $p < 0.05$, ** significant at $p < 0.01$, and if there is no sign, it is not significant.

It can be seen from Figure 2 that, before physiological maturity, the grain dehydration rate of hybrids of different ages show an increasing trend with the increase in density. The grain dehydration rate in the 2000s and 2010s hybrids was higher than that of the 1970s and 1980s. After physiological maturity, with the increase in density, except for the 1990s hybrids, the grain dehydration rate increased significantly, and the grain dehydration rate for the 1990s to 2010s hybrids was higher than that of the 1970s and 1980s.

By means of two years' data analyses, it was found that, before physiological maturity, from a low to medium density, the grain dehydration rate could be increased by 1.27–5.09%, and the grain dehydration rate was increased by 0.029, 0.044, 0.011, 0.030 and 0.019%/d, respectively. From a medium to high density, the grain dehydration rate could be increased by 1.64–11.12%, and the grain dehydration rate of the hybrids from various ages was increased by 0.101, 0.015, 0.022, 0.064 and 0.050%/d, respectively. The improvement effect of the 1970s hybrids was significant in two years, indicating that, before physiological maturity, the 1970s were sensitive to high density. The dehydration rate of other hybrids was not sensitive to density.

After physiological maturity, from a low to medium density, the seed dehydration rate could be increased by 0.00~14.55%, and increased by 0.050, 0.040, 0.000, 0.009 and 0.025%/d, respectively. From a medium to high density, the grain dehydration rate increased by 1.85~12.76%, and increased by 0.037, 0.008, 0.022, 0.015 and 0.058%/d, respectively. Except for the 1990s hybrids, the grain dehydration rate increased significantly with the change in

density. The grain dehydration rate of the 2010s hybrids obviously increased under a high density condition.

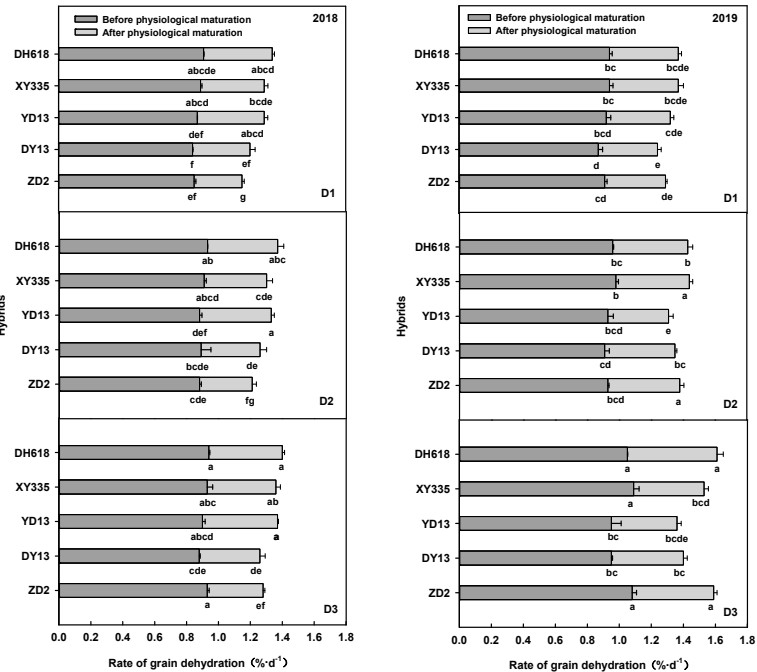

**Figure 2.** Changes in grain dehydration rate before and after physiological maturity of different cultivars at different densities in 2018 and 2019. Different letters on the graph indicate significant differences at $p < 0.05$.

### 3.3. Effects of Planting Density on Stay-Green of Different Hybrids

3.3.1. Effect of Relative GLAD after Silking

Relative GLAD is an important index to measure the stay-green of hybrids, and a higher relative GLAD shows that a better stay-green. As shown in Figure 3, with the development of the decade, the greenness of hybrids increased, and the greenness of hybrids in the 2000s and 2010s was better than that in the 1970s–1990s. As the density increased, the relative GLAD gradually decreased. Compared to the relative GLAD, the five hybrids of a low and medium density decreased by 9.74, 2.95, 4.85, 5.04 and 0.97% ·d plant$^{-1}$, and the planting density of the 2010s hybrids was still stable compared to the relative GLAD, maintaining a good greenness, and the decrease in hybrids from the 1970s to 2000s reached a significant level. The five hybrids of a medium and high density decreased by 4.55, 3.28, 2.03, 3.55 and 5.90% ·d plant$^{-1}$, respectively, while the hybrids from the 2010s (DH618), decreased significantly compared to the relative GLAD at a high planting density.

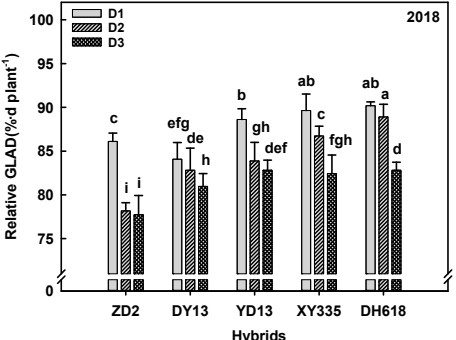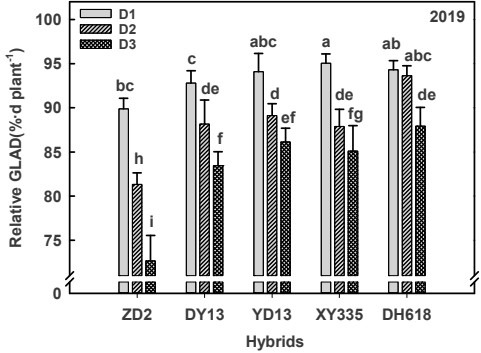

**Figure 3.** Changes in relative GLAD under different densities in 2018 and 2019. Different letters on the graph indicate significant differences at $p < 0.05$.

3.3.2. Influence on Chlorophyll Relative Content (SPAD Value)

As can be seen from Figure 4 that the SPAD value of leaves after the silking of hybrids of different ages was significantly higher in the 2010s than in the 1970s–2000s. With the development of the growth process, the SPAD value decreased significantly, and the decline was more rapid after 30 days of silking. From 30 to 60 days after silking, the SPAD value of hybrids decreased by 15.6, 14.9, 14.4, 14.1 and 14.3, respectively, under a low density. Under a medium density, the values of SPAD decreased by 18.2, 17.5, 15.7, 15.3 and 14.7, respectively, while under high density, they decreased by 20.8, 21.0, 17.7, 16.1 and 16.0, respectively.

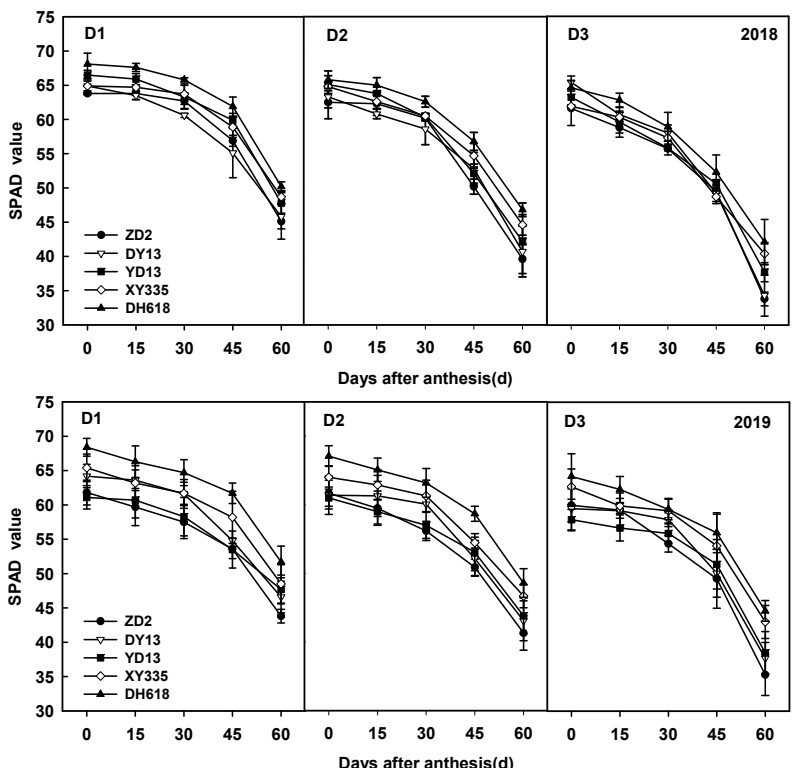

**Figure 4.** Variation of SPAD value under different densities in 2018 and 2019.

In the same period, the value of SPAD decreased with the increase in density. The SPAD values of low- to medium-density hybrids decreased by 4.5, 4.3, 3.4, 3.0 and 2.7, respectively, and the SPAD values of medium- to high-density hybrids decreased by 5.7, 5.9, 5.4, 4.4 and 4.6, respectively, at 60 days after silking. The SPAD value of the 1970s–1980s hybrids began to rapidly decline at a low to medium density, while the SPAD value of the 1990–2010s hybrids decreased significantly at a medium to high density.

*3.4. Effects of Increasing Planting Density on Dehydration Characteristics of Ear Organs in Different Hybrids*

3.4.1. Effect of Increasing Planting Density on the Dehydration Rate of Bracts

As can be seen from Figure 5, with the advancing of eras, the dehydration rate of bracts before and after physiological maturity shows a trend of first decreasing and then increasing; the dehydration rate of the bracts before and after physiological maturation in the 1980s was the slowest. With the increase in density, the dehydration rate of the bract before and after physiological maturity showed an increasing trend.

Before physiological maturity, the dehydration rate of bracts increased by 6.99~14.34% from a low to medium density, and increased by 0.072, 0.115, 0.061, 0.095 and 0.137%/d, respectively; for hybrids from the 1980s, 2000s and 2010s, the increase rate was higher, and the improvement effect was significant. From a medium to high density, the dehydration rate of the bracts increased by 12.83–30.11%; the dehydration rate of the bracts increased by

0.204, 0.181, 0.279, 0.163 and 0.154%/d, respectively; and the dehydration rate of the bracts increased by more than 10% in all the hybrids.

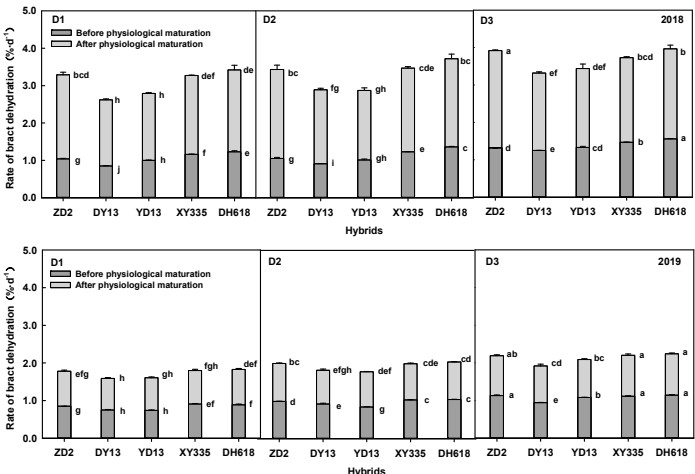

**Figure 5.** Dehydration rate of bracts of different hybrids under different densities in 2018 and 2019. Different letters on the graph indicate significant differences at $p < 0.05$.

After physiological maturity, the dehydration rate of the bracts increased by 4.99–10.66% from a low to medium density, which was increased by 0.105, 0.139, 0.067, 0.098 and 0.115%/d for hybrids of different ages, respectively. The improvement effect of the 2000s and 2010s hybrids was significant in two years of the experiment. From a medium to high density, the dehydration rate of the bracts could be increased by 4.50–11.30%. The dehydration rate of the bracts of hybrids of different ages was increased by 0.139, 0.085, 0.158, 0.075 and 0.076%/d, respectively. Except for the 1980s, the dehydration rate of the bracts of other hybrids was increased by more than 10%, showing a significant improvement effect.

3.4.2. Effect of Increasing Planting Density on the Dehydration Rate of Cobs

Figure 6 shows that, under the same planting density, the cob dehydration rate first decreased and then increased before and after physiological maturity with age, and the cob dehydration rate of the 1990s hybrids was slow. With the increase in density, the cob dehydration rate decreased.

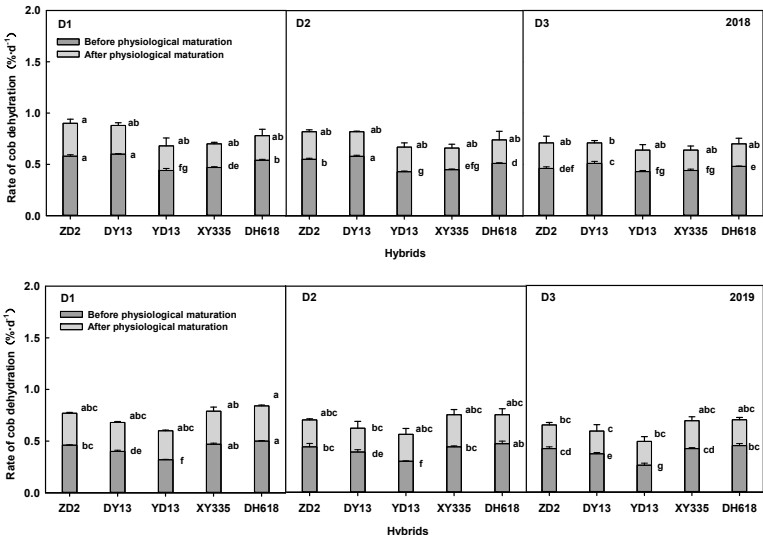

**Figure 6.** Dehydration rate of cobs of different hybrids under different densities in 2018 and 2019. Different letters on the graph indicate significant differences at $p < 0.05$.

Before physiological maturity, the cob dehydration rate decreased by 3.23–5.96% from a low to medium density, and the cob dehydration rate decreased by 0.024, 0.016, 0.018, 0.024 and 0.031%/d for hybrids of different ages, respectively. However, the decrease in the cob dehydration rate was not significant. From a medium to high density, the cob dehydration rate decreased by 3.05~11.09%. The cob dehydration rate decreased by 0.055, 0.043, 0.020, 0.014 and 0.028%/d, respectively. The decrease rate was low in the 1990s and 2000s, and the cob dehydration rate did not decrease significantly.

After physiological maturity, the cob dehydration rate decreased by 4.56~15.64% from a low to medium density. The cob dehydration rate decreased by 0.049, 0.045, 0.012, 0.018 and 0.041%/d, respectively, but the decrease rate was not significant in the 1990s and 2000s. From a medium to high density, the cob dehydration rate decreased by 5.41–10.46%, and the cob dehydration rate decreased by 0.025, 0.024, 0.026, 0.026, 0.014%/d, respectively; for hybrids from the 2010s, the decreasing range was the lowest, and the cob dehydration rate of hybrids of various ages did not reach significance.

### 3.5. Path Analysis of Ear Organ Dehydration Rate and Grain Water Index

Since the dehydration rates of bracts and cobs are not consistent in response to density, their effects on grain dehydration characteristics need further analysis. According to the path analysis results presented in Table 8, the dehydration rate of the bracts before physiological maturity had significant ($p \leq 0.05$) direct negative and direct positive effects on the grain moisture content at physiological maturity and harvest stages, and on the grain dehydration rate before and after physiological maturity, respectively. Bracts had little effect on the grain water index after physiological maturity. The cob dehydration rate before physiological maturity had a significant ($p \leq 0.05$) direct negative effect on the grain moisture content at physiological maturity and harvest stages, but had a weak positive effect on the grain dehydration rate before and after physiological maturity. The cob dehydration rate after physiological maturity had a weak positive effect on the grain moisture content at physiological maturity and harvest stages, while it had a direct negative effect on the grain dehydration rate before and after physiological maturity.

**Table 8.** Path analysis of dewatering rate and grain water index of bract and cob before and after physiological maturity.

| Dependent Variable | Index | Correlation Coefficient | Direct Path Coefficient | Indirect Path Coefficient | | | |
|---|---|---|---|---|---|---|---|
| | | | | x1 | x2 | x3 | x4 |
| Y1 | x1 | −0.921 | −0.846 | - | −0.051 | 0.043 | −0.068 |
| | x2 | −0.857 | −0.061 | −0.704 | - | −0.063 | −0.029 |
| | x3 | −0.120 | −0.309 | 0.118 | −0.012 | - | 0.083 |
| | x4 | 0.379 | 0.145 | 0.399 | 0.012 | −0.177 | - |
| Y2 | x1 | −0.843 | −0.974 | - | 0.102 | 0.068 | −0.039 |
| | x2 | −0.804 | 0.122 | −0.811 | - | −0.099 | −0.016 |
| | x3 | −0.278 | −0.486 | 0.136 | 0.025 | - | 0.047 |
| | x4 | 0.238 | 0.082 | 0.459 | −0.024 | −0.279 | - |
| Y3 | x1 | 0.917 | 0.683 | - | 0.201 | 0.006 | 0.027 |
| | x2 | 0.812 | 0.242 | 0.568 | - | −0.009 | 0.011 |
| | x3 | −0.123 | −0.044 | −0.095 | 0.049 | - | −0.033 |
| | x4 | −0.452 | −0.058 | −0.322 | −0.048 | −0.025 | - |
| Y4 | x1 | 0.784 | 1.226 | - | −0.490 | −0.012 | 0.060 |
| | x2 | 0.475 | −0.589 | 1.020 | - | 0.018 | 0.025 |
| | x3 | −0.278 | 0.086 | −0.171 | −0.120 | - | −0.073 |
| | x4 | −0.539 | −0.127 | −0.578 | 0.117 | 0.049 | - |

Y1: moisture content in physiological maturity; Y2: moisture content at harvest; Y3: rate of grain dehydration before maturation; Y4: rate of grain dehydration after maturation; x1: rate of bract dehydration before maturation; x2: rate of bract dehydration after maturation; x3: rate of cob dehydration before maturation; x4: rate of cob dehydration before maturation.

The direct negative effect of bract dehydration rate on grain moisture content at the physiological maturity stage (Px1-Y1 = −0.846) and grain moisture content at the harvest stage (PX1-Y2 = −0.974) was much greater than that of the cob (Px3-Y1 = −0.309, px3-Y2 = −0.486). The positive and direct effects of px3-Y3 = 0.683 and Px1-Y4 = 1.226 on the dehydration rate of grain before and after physiological maturity were much greater than those of the cob (px3-Y3 = −0.044 and Px3-Y4 = 0.086). The negative direct effect of bract dehydration rate on grain moisture content at physiological maturity (PX2-Y1 = −0.061) was less than that of cob (PX4-Y1 = 0.145), and the positive direct effect on grain moisture content at harvest (PX2-Y2 = 0.122) was greater than that of the cob (PX4-Y2 = 0.082). The positive and direct effects of bract dehydration rate on grain dehydration rate before physiological maturity (PX2-Y3 = 0.242) were greater than that of cob (PX4-Y3 = −0.058), and the negative direct effects of bract dehydration rate on grain dehydration rate after physiological maturity (PX2-Y4 = −0.589) were greater than that of the cob (PX4-Y4 = −0.127).

## 4. Discussion

There were different opinions about the effect of density on the dehydration characteristics of maize grains. Xu et al. [27] conducted a study at the planting density of 45,000–75,000 plants ha$^{-1}$, and showed that, with the increase in population density, the dehydration rate decreased, and the grain moisture content showed a gradually increasing trend at physiological maturity and harvest stages. Feng et al. [28] conducted a study at a planting density of 60,000–105,000 plants ha$^{-1}$ and showed that density had no significant effect on the average dehydration rate and water content of grains before physiological maturity, but had a great effect on the average dehydration rate of grains at later physiological maturity. Yu et al. [29] showed that, when the planting density increased from 82,500 plants ha$^{-1}$ to 112,500 plants ha$^{-1}$, the dehydration rate of maize could increase by 3.99–6.33%. With the change in planting density, the growth micro-environment of maize was changed, the agronomic traits of maize were changed, and the dewatering performance of maize grain was affected. In this experiment, the grain dehydration rate of different hybrids increased with the increase in planting density, and increased more considerably after physiological maturity. The difference may be due to dry climate conditions in the test area.

The grain dehydration rate significantly increased for the 1970s hybrid before physiological maturity, and significantly increased for the 2010s hybrid after the physiological maturity in high-density conditions. The difference between hybrids was due to their own characteristics for maize production demands in years of release. In the 1970s, maize breeding in China pursued resistance to hybrids of common diseases and wide adaptability. In the 1980s and 1990s, China pursued compact plant type, large ear and high yield. In 2000 and 2010, it began to pursue high yield, stable yield, high quality, multi-resistance and wide adaptability [26].

Previous studies found that plant height, leaf area index, number of green leaves at filling stage, ear length, yield per ear and other plant physiological and agronomic traits were closely related to grain dehydration performance [30]. Zhang et al. [12] showed that the maize grain water content at physiological maturity increased significantly with the increase in LAI. Among the agronomic traits, bract characteristics had a significant impact on the grain moisture content [31]. Crane et al. [32] reported that bract length was negatively correlated with the grain moisture content. Hicks et al. [22] showed that the bract tightness was correlated with the natural dehydration rate of grains in the field. Cross et al. [33] confirmed that more bract layers and bract biomass were not conducive to grain dehydration.

On the one hand, after densification, the relative GLAD and leaf SPAD values of maize per plant significantly decreased, the plant stay-green significantly decreased and the plant accelerated senescence. On the other hand, the dehydration of the bract before and after physiological maturity was significantly increased, and these two factors jointly promoted the increase in grain dehydration rate before the physiological maturity of the 1970s hybrids. The significantly increased grain dehydration rate of the 2010s hybrids after physiological

maturity under high-density conditions was attributed to the significantly decreased plant green retention performance of the 2010s hybrids under high-density conditions, and the significantly increased bract dehydration before and after physiological maturity.

Thus, we found that the hybrids from the early 1970s and the modern hybrids from the 2010s increased in grain dehydration rate significantly after being added close together after the plant relative GLAD and leaf SPAD values fell dramatically, the plant stay-green degradation caused by plant rapid senescence, and the planting density increased the bract dehydration rate significantly; the two factors were promoted together. This suggested that, under the condition of high density, the selection of hybrids with significantly decreased plant stay-green and a significantly increased dehydration rate of ear bracts was beneficial to the loss of maize grain water. However, the limitation of this study was that the representative varieties were single. In subsequent studies, we can select three to five representative varieties in each era to further verify and find out more reasons for why, after the planting density increases, the acceleration of the grain dehydration rate is promoted, so as to enrich our selection criteria for the varieties suitable for machine harvest in the future to provide a basis for the breeding of maize varieties suitable for machine harvest, to achieve the high yield, high quality and high efficiency of maize, which provides a more abundant theoretical reference.

## 5. Conclusions

The response of grain moisture content and dehydration rate to increasing planting density was different among the maize hybrids. With the advancing of the eras, the grain dehydration rate before the physiological maturation of the hybrids increased after densification, and the grain dehydration rate of the hybrids in the 2000s and 2010s was higher than that in 1970s and 1980s. However, the grain dehydration rate after the physiological maturation of hybrids after densification showed an obvious increasing trend, except for the 1990s hybrids; the grain dehydration rate of the 1990s–2010s hybrids was higher than that of the 1970s and 1980s hybrids. The grain moisture content in the physiological maturity and harvest stages in the 1970s was significantly affected by the increase in planting density, and significantly decreased by 1.57 and 0.99 percentage points from the low- to medium-density physiological maturity stage, respectively. The grain water content of the 1980s hybrids at the physiological maturity and harvest stages was greatly affected the by high planting density, and decreased by 1.78 and 1.77 percentage points from a medium to high density, respectively. In the 1990s, the seed water content during physiological maturity was significantly affected by the planting density, which decreased by 1.14, 1.53, and 1.14, respectively, from a low to medium to high density. In the 2000s, the seed water content was significantly affected by a high planting density. The grain water content at the harvest stage was less affected by planting density when the low to medium density decreased by 1.88 percentage points, and the grain water content at the physiological maturity and harvest stages was not sensitive to the planting density. Further analysis showed that, after the planting density increased, the relative GLAD and leaf SPAD values of plants decreased, the stay-green of the plants worsened and the significant increase in the dehydration rate of bracts was an important reason to promote the decrease in grain moisture content and the increase in dehydration rate.

**Author Contributions:** X.Y., J.G. and H.G conceived and designed the experiments; H.G., X.W. and S.H. performed the experiments; H.G., X.Y. and X.W analyzed the data; X.Y., J.G. and D.M. critically revised it for important intellectual content; H.G. and X.Y wrote the paper. All authors have read and agreed to the published version of the manuscript.

**Funding:** This study was funded by the National Natural Science Foundation of China (Grant No. 31560360), Science and Technology Innovation Projects for High Yield and Efficiency of Grain (Grant No. 2017YFD0300804), National Maize Industrial Technology Systems (Grant No. CARS-02-50) and Crop Science Observation and Experiment Station in Loess Plateau of North China, Ministry of Agriculture, P.R. of China (Grant No. 25204120), and (KJXM2020001-06) The Key Program of Action Plan to Revitalize Inner Mongolia through Science and Technology (KJXM2020001).

**Institutional Review Board Statement:** Not applicable.

**Informed Consent Statement:** Not applicable.

**Data Availability Statement:** Not applicable.

**Conflicts of Interest:** The authors declare no conflict of interest.

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
