# Peer review of "Grain Dehydration Characteristics of Old and Modern Maize Hybrids and Their Response to Different Planting Densities"

_agronomy, doi:10.3390/agronomy12071640_

Round 1

Reviewer 1 Report

Nevertheless, one minor revision:

- check the orthography in the whole text: some words are attached with successive parenthesis (i.e. row 165 "time.MCh"); commas or dots are attached with successive words (i.e. row 466 "maturity.Tan"

Author Response

The revised version and reply are below.

Reviewer 2 Report

Thanks, dear, I saw editing after revision and accepted all (Last response) Kind Regards

Author Response

The revised version are below.

Reviewer 3 Report

The manuscript is improved and in present form, is more fluent, eazy to understand, with more clear massages.

Besides very few small corrections.

Author Response

The revised version are below.

This manuscript is a resubmission of an earlier submission. The following is a list of the peer review reports and author responses from that submission.

Round 1

Reviewer 1 Report

  1. The structure of the introduction is in need of improvement. There are too long sentences that make the sense of the text get lost, with wrong punctuation use as well. I advise the authors to properly read lines 88 to 106 to make them clearer. Nevertheless, the cited articles seem to support the central problem of the article. 
  2. in materials and methods is listed twice the fertilizer directions (rows 114-116 and rows 125-129). I ask the authors to clarify. 
  3. When the variables referring to Grain dehydratation characteristics are presented, the acronyms used in the introduction are not presented, making reading the article very difficult (rows 145-154 and rows 159-166)
  4. In the results and discussions, it is complicated to follow the discussion. Reference is made to the years (the 1970s, 1980s, 1990s, 2000s, 2010s) but not to the varieties reported in materials and methods, which instead appear in all tables and figures. I advise the authors to fix these references in order to facilitate clarity.

Reviewer 2 Report

Your manuscript need more details to improve it.

  • The title of this manuscript must be improved (not use effect word, that better).
  • Abstract must contain these elements (principle objectives, methods used, place, region, experimental design……, principle results, main conclusion), these elements not cleared, it must revise.
  • In introduction, it must be contained maize importance in your country and in the world (Short sentence).
  • The aim of the study not clear in introduction.
  • Your aim not clear.
  • You can present you meteorological data in Table, also, soil analysis in Table.
  • Put the pedigree of the used varieties (Table).
  • You can describe your experimental design better than, determine main plot (factor and its treatments), subplots (factor and treatments).
  • Can you put Plant density (spacing for row and plants).
  • What do you mean cell in materials and methods part.
  • How to use two method (LSD and Duncan’s method).
  • SAS program an SPSS, need Ref.
  • I thank, can you calculate LAD or any methods for compare means among treatments using SAS prog.
  • Results and discussion; this part need more details such as it must present the main effect.
  • Can you delete ANova table.